# The Value of Rapid Antigen Tests for Identifying Carriers of Viable SARS-CoV-2

**DOI:** 10.3390/v13102012

**Published:** 2021-10-06

**Authors:** Elena V. Shidlovskaya, Nadezhda A. Kuznetsova, Elizaveta V. Divisenko, Maria A. Nikiforova, Andrei E. Siniavin, Daria A. Ogarkova, Aleksandr V. Shagaev, Maria A. Semashko, Artem P. Tkachuk, Olga A. Burgasova, Vladimir A. Gushchin

**Affiliations:** 1N.F. Gamaleya National Research Center for Epidemiology and Microbiology, Ivanovsky Institute of Virology, Ministry of Health of the Russian Federation, 123098 Moscow, Russia; lenitsa@gmail.com (E.V.S.); nadyakuznetsova0@gmail.com (N.A.K.); elizaveta.divisenko@yandex.ru (E.V.D.); marianikiforova@inbox.ru (M.A.N.); andreysi93@ya.ru (A.E.S.); DashaDv1993@gmail.com (D.A.O.); maria.a.semashko@gmail.com (M.A.S.); artem.p.tkachuk@gmail.com (A.P.T.); olgaburgasova@mail.ru (O.A.B.); 2Department of Molecular Neuroimmune Signalling, Shemyakin-Ovchinnikov Institute of Bioorganic Chemistry, Russian Academy of Sciences, 117997 Moscow, Russia; 3Moscow Healthcare Department, 127006 Moscow, Russia; alexandrshagayev@gmail.com; 4Department of Infectious Diseases with the Courses of Epidemiology and Phthisiology, Peoples Friendship University of Russia (RUDN) University, 117198 Moscow, Russia; 5Department of Virology, Lomonosov Moscow State University, 119991 Moscow, Russia

**Keywords:** SARS-CoV-2, COVID-19, viable virus, rapid antigen tests

## Abstract

The search for effective methods to detect patients who excrete a viable virus is one of the urgent tasks of modern biomedicine. In the present study, we examined the diagnostic value of two antigen tests, BIOCREDIT COVID-19 Ag (RapiGEN Inc., Anyang, Korea) and SGTI-flex COVID-19 Ag (Sugentech Inc., Cheongju, Korea), for their diagnostic value in identifying patients who excrete viable SARS-CoV-2. As part of the study, we examined samples from 106 patients who had just been admitted to the hospital and who had undergone quantitative RT-PCR and assessment of viability of SARS-CoV-2 using cell culture. Assessment of the tests’ value for detecting samples containing viable virus showed high sensitivity for both tests. Sensitivity was 78.6% (95% CI, from 49.2% to 95.3%) for SGTI-flex COVID-19 Ag and 100% (95% CI, from 76.8% to 100%) for Biocredit COVID-19 Ag. The specificity of rapid tests was significantly higher than that of RT-PCR and was 66.3% (95% CI, from 55.7% to 75.8%) and 67.4% (95% CI, from 56.8% to 76.8%) for SGTI-flex COVID-19 Ag and Biocredit COVID-19 Ag versus 30.4% (95% CI, from 21.3% to 40.9%) obtained for PCR. Thus, for tasks of identifying viable SARS-CoV-2 during screening of conditionally healthy people, as well as monitoring those quarantined, rapid tests show significantly better results.

## 1. Introduction

The SARS-CoV-2 virus pandemic has been a major global problem for over a year. The main problem in the monitoring and surveillance of SARS-CoV-2 is the ability of the virus to spread from asymptomatic patients several days before any symptoms occur [1,2]. Moreover, the contribution to the transmission of the virus from asymptomatic patients and from patients before the onset of symptoms is a significant problem both for the spread of the virus and for accounting for COVID-19 cases [3]. The identification of carriers of a viable virus is critical to prevent the infection of other people and the further spread of the virus [4]. In addition, the emergence of new variants of SARS-CoV-2, including variants of concern, requires further assessment of the ability of existing kits to identify patients shedding viable virus [5,6].

Rapid antigen tests, which have recently become widespread for the diagnosis of COVID-19, in contrast to PCR, detect SARS-CoV-2 antigens, which, similar to RNA, comprise viral particles and are produced in infected cells during the life cycle of the virus. The disadvantage of antigen tests is their lower sensitivity compared to RT-PCR. According to the results of a Cochrane meta-analysis, sensitivity varies greatly: average sensitivity is 56.2% (95% CI, from 29.5% to 79.8%), and average specificity is 99.5% (95% CI, from 98.1% to 99.9%; based on eight experiments in five studies on 943 samples) [7]. At the same time, the definitive advantage of rapid antigen tests is the significantly less laborious process and the ease of learning the procedure, allowing the test at home to be used as point of care (POC) testing. The time to obtain the result can be as low as 5 min, which is also crucial. Moreover, rapid antigen tests are not susceptible to contamination with amplification products, characteristic of nucleic acid analysis methods, which reduces the likelihood of a false positive result.

In this paper, we describe results of the study of two rapid antigen tests, BIOCREDIT COVID-19 Ag (RapiGEN Inc., Anyang, Korea) and SGTI-flex COVID-19 Ag (Sugentech Inc., Cheongju, Korea), for their value in identifying patients excreting viable SARS-CoV-2. As part of the study, we examined samples from 106 patients who had just been admitted to the hospital and who had undergone two rapid tests, quantitative RT-PCR and viability assessment of SARS-CoV-2 using susceptible cell culture 293T/ACE2. Samples were collected from 25 January 2021 to 8 February 2021 at an infectious diseases hospital in Moscow. Additionally, in this work, we evaluate the ability of the selected antigenic tests to detect different variants of the virus.

## 2. Materials and Methods

### 2.1. Patients

The study included patients with suspected COVID-19 admitted to the hospital on day 2–10 from the onset of symptoms (fever, dry cough, chest pain and discomfort, shortness of breath, loss of smell and taste) and with CT signs of lung damage. Study was approved by the local ethic committee of the Moscow First Infectious Diseases Hospital (the Protocol #2 dated 22 January 2021). All participants signed the written informed consent to allow usage of nasal swab samples for research purposes.

### 2.2. Sample Collection and Transportation

Using sterile swabs and observing the necessary safety precautions, nurses of the hospital collected three nasopharyngeal samples from each patient, two of which were used for antigen testing. The first sample was transferred to tubes with 1 mL of phosphate-buffered saline (PBS) and then used for RT-PCR and virus isolation. On the next day (after obtaining results of RT-PCR), participants were tested by antigenic tests. All collected materials were transferred to the reference center for coronavirus infection of the Gamaleya Research Institute of Epidemiology and Microbiology of the Ministry of Health of Russia, cooled down to +4 °C within 2 h after collection.

Blood samples were collected by venipuncture to vacutainers with clot activator and shipped to the laboratory at +4 °C. Centrifugation at 3000 rpm for 10 min was applied to obtain serum, which was further aliquoted and stored at −30 °C.

### 2.3. Anti-Nucleocapsid and Anti-RBD IgG Antibody Detection

For antibody detection, we used recombinant receptor-binding domain fragment of S1 SARS-CoV-2 spike protein (Cat.No. 8COV1, HyTest, Moscow, Russia) and recombinant nucleoprotein (Cat.No. 8COV3, HyTest, Moscow, Russia), expressed in eukaryotic cells. To perform anti-RBD and anti-Nc ELISAs, 96-well high binding plates (Costar 2592, Corning, New York, NY, USA) were coated overnight with 100 μL of 1 μg/mL recombinant protein solution in PBS. The next day, the plates were blocked for 2 h at room temperature with a blocking buffer containing 0.5% casein. Serum samples were diluted 1:100 with universal ELISA buffer S011 (XEMA, Moscow, Russia). Sera of PCR + reconvalescent were used as a positive control, and a pool of pre-COVID samples, collected in 2019, was used as a negative control. ELISA plates with 100 μL of diluted samples were incubated 1 h at 37 °C and washed 3 times with PBS containing 0.1% Tween-20. After washing, wells were incubated with 100 μL of HRP-conjugated anti-human IgG (Novex A18823, USA) for 1 h at 37 °C and washed 6 times. After adding 100 μL of the HRP substrate solution, containing 3,3′,5,5′-Tetramethylbenzidine (R055, XEMA, Moscow, Russia) per well, the color reaction was developed for 10 min at room temperature and then stopped by 10% HCl. Optical density (OD) was measured at 450 nm using Multiscan FC (Thermo Scientific, Waltham, MA, USA).

### 2.4. Antigen Testing

Antigen testing was done immediately after sample collection in accordance with the manufacturer’s instructions directly at the patient’s bedside. Testing was done using two commercially available rapid tests for the detection of SARS-CoV-2 antigen: BIOCREDIT COVID-19 Ag (RapiGEN Inc., Anyang, Korea) and SGTI-flex COVID-19 Ag (Sugentech Inc., Daejeon, Korea).

### 2.5. SARS-CoV-2 Testing by Quantitative RT-PCR

All collected samples were tested immediately after transportation. PCR amplification was carried out using a one-step “SARS-CoV-2 FRT” commercial kit with catalog number EA-128 (bought from N.F. Gamaleya NRCEM, Moscow, Russia). According to manufacturer’s information, “SARS-CoV-2 FRT” kit allows for amplifying a fragment from the 5′ end region encoding the NSP1 gene (approximately 450 to 650 nt bases upstream the 5′ end of SARS-CoV-2 viral genome). Briefly, the conditions of the one-step RT-qPCR reaction were as follows: 50 °C for 15 min, 95 °C for 5 min, followed by 45 cycles of 95 °C for 10 s and 55 °C for 1 min. The number of copies of viral RNA was calculated using a standard curve generated by amplification of plasmid cloned DNA template fragment encoding 450 to 650 nt bases upstream the 5′ end of SARS-CoV-2 viral genome.

### 2.6. Virus Isolation

Isolation of the SARS-CoV-2 virus was performed using 293T/ACE2 cell line (with stable expression of the human ACE2 receptor). Cells were cultured in DMEM medium (PanEco, Moscow, Russia) containing 10% FBS (HyClone, Logan, UT, USA), 1% L-glutamine, and 1% penicillin/streptomycin. A 96-well plate was used for the experiment. For this, nasopharyngeal secretion (100 μL) from COVID-19 patients was added to cells. Plates were incubated for 5 days. Virus-induced cytopathic effect (CPE) was then assessed. Additionally, for samples with CPE, real-time PCR was performed to confirm that CPE was caused by SARS-CoV-2 and not by other infectious agents that can cause CPE.

### 2.7. Statistical Treatment of Results

All data were statistically treated using the methods available in different R packages. McNemar’s test with Edwards continual correction was used to compare the two different tests. Fisher’s exact test was used to analyze unrelated qualitative data. Quantitative indicators were checked for normal distribution using the Shapiro–Wilk test. Quantitative comparison of groups was done using the Mann–Whitney test. Confidence intervals for specificity and sensitivity, as well as confidence intervals for proportions, were calculated using binomial distribution with the help of Clopper–Pearson method.

## 3. Results

### 3.1. Study Design

To investigate the ability of rapid antigen tests to identify patients who excrete viable SARS-CoV-2, it was planned to include primarily patients with suspected COVID-19 newly admitted to the hospital. Participants were admitted to the hospital on 2 to 10 days from the onset of symptoms. The main symptoms were fever, dry cough, chest pain and discomfort, shortness of breath, and loss of smell and taste. All included patients had CT signs of lung damage. The study included 106 patients aged 28 to 95 years (mean age 67.67), including 53 women (mean age 68.45) and 53 men (mean age 66.89) (Appendix A Table A2).

To confirm infection, nasopharyngeal samples were examined by quantitative RT-PCR. In addition, we collected convalescent sera from the patients about three weeks after onset of the symptoms to evaluate antibody response by ELISA. The fact of having recovered or not being infected by SARS-CoV-2 was verified for 99 patients. Seven patients were discharged from the hospital earlier than three weeks after the onset of symptoms, and we were unable to obtain their serum samples.

### 3.2. Antigen Tests Detection of SARS-CoV-2 Variants

To evaluate the ability of the antigen tests to detect new variants of SARS-CoV-2, we carried out additional testing isolates of the following virus variants: Alpha lineages B.1.1.1 (EPI_ISL_421275), B.1.1 (EPI_ISL_1710865), B.1.1.7 (EPI_ISL_1710866); Beta lineage B.1.351 (EPI_ISL_1257814); and Delta lineage B.1.617.2. In addition, we tested one isolate of SARS-CoV-2 virus with E484K and S494P mutations in the RBD domain (EPI_ISL_2296305).

Sensitivity was about 25–58% for all virus variants. There was no significant difference between the analytical characteristics of the tests in all studied virus variants (Table 1, Appendix A Table A1). We have also not seen differences between sensitivity for each variant.

### 3.3. Analytical Characteristics of Antigen Tests Compared to RT-PCR

The result of PCR tests of swabs from 106 patients was positive in 73.58% patients (78 people). The viral load was determined for all positive samples (Appendix A Table A2, Table A3 and Table A4), which ranged from 88 to 3.5 × 10^8^ copies/mL (median 4.6 × 10^4^). The SGTI-flex COVID-19 Ag rapid test identified 41 of 78 positive samples. Sensitivity of SGTI-flex COVID-19 Ag was 52.56% (95% CI, from 40.9% to 63.99%), and specificity was 96% (95% CI, from 81.7% to 99.9%) (Table 2). In turn, the Biocredit COVID-19 Ag rapid test identified 44 of 78 positive samples. For the Biocredit COVID-19 Ag test, sensitivity was 56.41% (95% CI, 44.7–67.6%) and 100% (95% CI, 87.7–100%), respectively. There were no significant differences between the analytical characteristics of the tests (*p* = 0.8026, McNemar’s test with Edwards continuity correction).

For samples with a higher viral load, the sensitivity of tests was higher (Appendix A Table A2) and, starting from a viral load of 1.02 × 10^5^ (log10 = 5.0086) for SGTI-flex COVID-19 Ag and 4.74 × 10^4^ (log10 = 4.6758) for RapiGen Biocredit COVID-19 Ag, did not differ statistically from the results of PCR tests at a significance level of 0.01 (*p* = 0.01333, McNemar’s test with Edwards continuity correction). The *p*-value only increased with further increase in viral load.

To determine the analytical threshold of sensitivity with respect to the antigen in virions of the culture fluid, we conducted a model experiment when the culture fluid with a known virus titer was used to assess the analytical sensitivity of rapid tests in the range from 10^2^ to 10^8^ copies/mL, using an interval of one order of magnitude. Both tests showed the detection limit at a virus titer of 10^6^ copies/mL (10^5^ copies/test), which corresponded to 4 × 10^5^ TCID50/mL or 4 × 10^4^ TCID50/test. There was no statistically significant difference in analytical characteristics depending on the day from the onset of the disease (*p*-value = 0.2356 for SGTI-flex COVID-19 Ag, *p*-value = 0.8581 for RapiGen Biocredit COVID-19 Ag with Fisher’s exact test), which is not surprising given that there were high-load patients on each day (Appendix A Table A4).

### 3.4. Analytical Characteristics of Antigen Tests for Viability

All samples were evaluated for viability of SARS-CoV-2 virus using a sensitive cell culture. Viability was assessed using the 293 T/ACE2 cell line with stable expression of the human ACE2 receptor. For samples with a cytopathogenic effect (CPE), RT-PCR was performed to confirm that the CPE was caused by SARS-CoV-2 and not by other infectious agents. Comparison of groups of samples with viable and non-viable viral load measured by quantitative PCR showed a significant difference (*p* < 0.0001, *p*-value calculated using the Mann–Whitney test) (Figure 1).

Viability was shown only by samples with a viral load of 7.3 × 10^4^ (copies/mL) and higher (Figure 1 and Table 3). However, not all samples with such a load remained viable. Rapid tests were able to give a positive result on samples with median values of 1.03 × 10^6^ and 1.09 × 10^6^ (copies/mL) for Biocredit COVID-19 Ag and SGTI-flex COVID-19 Ag, respectively. However, a positive result was also obtained for a number of samples with a load below 10^4^ (copies/mL). It is not clear whether this is related to how the material was collected for three different tests or the presence in some biological samples of a disproportionate amount of antigen in relation to viral RNA.

Overall, out of 106 samples, a viable virus was detected in 14 patients, representing 13.2% of all participants and 17.9% of RT-PCR-positive participants. Using these samples, SGTI-flex COVID-19 Ag gave 11 (78.6%) positive results, while Biocredit COVID-19 Ag gave 14 (100%) positive results (Table 4). The sensitivity of antigen tests was 78.6% (95% CI, from 49.2% to 95.3%) for SGTI-flex COVID-19 Ag and 100% (95% CI, 76.8–100%) for Biocredit COVID-19 Ag. Specificity was 66.3% (95% CI, 55.7–75.8%) and 67.4% (95% CI, 56.8–76.8%) for SGTI-flex COVID-19 Ag and Biocredit COVID-19 Ag, respectively. For RT-PCR, the sensitivity was 100% (95% CI, 76.8–100%), while the specificity was 30.4% (95% CI, 21.3–40.9%).

Comparing analytical characteristics of the tests for viability of the virus, it can be observed that the rapid tests are indistinguishable from each other (*p*-value = 0.4533, McNemar’s test with Edwards continuity correction), but they differ significantly from RT-PCR (*p*-value = 2.546 × 10^−6^ for SGTI-flex COVID-19 Ag, *p*-value = 1.519 × 10^−8^ for Biocredit COVID-19 Ag, McNemar test with Edwards continuity correction).

As the viral load increased, the sensitivity of the tests for viral viability increased and, starting from a viral load of 6.27 × 10^4^ (log10 = 4.7973) for SGTI-flex COVID-19 Ag and 5.11 ×10^5^ (log10 = 5.7084) for RapiGen Biocredit COVID-19 Ag, did not statistically significantly differ from the possibility of successful isolation of a viable virus at a significance level of 0.01 (*p* = 0.01529 and *p* = 0.01333, respectively).

There were no statistical differences in the analytical characteristics of rapid tests depending on the day of the disease (*p*-value = 0.5292 for SGTI-flex COVID-19 Ag, 0.4108 for RapiGen Biocredit COVID-19 Ag using Fisher’s exact test). At the same time, the number of cases of successful isolation of a viable virus also did not statistically differ at different time intervals of the disease, which is probably due to the limited follow-up period and the use of only one sample from each patient.

## 4. Discussion

The prompt identification of carriers of viable SARS-CoV-2 virus and their timely isolation from society is the most important step in containing a pandemic. Mass screening, in turn, is limited by the access of the population to laboratory testing, which in the case of PCR is limited by the capabilities of laboratories. In this regard, many states are adopting new approaches, whereby samples are pooled to increase the amount of PCR tests [8,9,10,11]. An alternative to using PCR laboratories is to use rapid antigen tests, including for home use.

We examined samples from 106 patients admitted to a hospital in Moscow who had experienced their first symptoms no earlier than 10 days before. All samples, in addition to being used for the two tests, were also used for quantitative PCR and assessment of viral viability using cell culture. Of 106 samples, 78 (73.58%) were PCR-positive. This is significantly higher than previously published data with similar clinical settings [12]. This is probably due to a difference in design, as, for the purposes of this study, we searched for patients who had just been admitted to the hospital, while other studies did not make such a distinction [13,14,15].

Data of quantitative PCR are of particular interest. The viral load was determined for all PCR-positive samples, and it was found that it varied greatly, from 88 to 3.5 × 10^8^ copies/mL (median 4.6 × 10^4^). In terms of viral load, samples that showed viability were significantly different from the rest (*p* < 0.0001, the p-value was calculated using the Mann–Whitney test). For all 14 samples that showed viability, the viral load was at least 7.3 × 10^4^ (copies/mL). Additionally, although not all samples with a similar load or higher showed viability, it is important that a cut-off quantitative threshold, after which the probability of the virus remaining viable is greatly reduced, can be established experimentally.

A standard evaluation of the analytical characteristics of rapid tests showed that, relative to RT-PCR, their diagnostic characteristics were close to the mean values published for the Cochrane meta-analysis [7]. For the SGTI-flex COVID-19 Ag test, sensitivity was 52.56% (95% CI, 40.9–63.99%), and specificity was 96.4% (95% CI, 81.7–99.9%). For the Biocredit COVID-19 Ag test, sensitivity and specificity were 56.41% (95% CI, 44.7–67.6%) and 100% (95% CI, 87.7–100%), respectively. No significant differences were found between tests (*p* = 0.8026). Analysis of the value of tests for detecting samples containing viable virus showed that both tests are highly sensitive. Sensitivity was 78.6% (95% CI, 49.2–95.3%) and 100% (95% CI, from 76.8% to 100%) for SGTI-flex COVID-19 Ag and Biocredit COVID-19 Ag. The control method (RT-PCR) had sensitivity of 100% (95% CI, from 76.8% to 100%). In turn, the specificity of rapid tests was significantly higher than that of RT-PCR and was 66.3% (from 55.7 to 75.8%) and 67.4% (from 56.8% to 76.8%) for SGTI-flex COVID-19 Ag and Biocredit COVID-19 Ag versus 30.4% (from 21.3% to 40.9%) obtained for PCR. Statistically, the results of the antigen tests used in the study are indistinguishable from each other (*p*-value = 0.4533) but differ significantly from RT-PCR (*p*-value = 2.546 × 10^−6^ for SGTI-flex COVID-19 Ag, *p*-value = 1.519 × 10^−8^ for Biocredit COVID-19 Ag). This means that rapid tests have significantly better results for the task of identifying viable SARS-CoV-2. It should be noted that the quality of sample collection from patients affects the results of the study.

In this study, there were 14 samples that showed viability. It appears important to investigate to what extent the results of this study will correlate with data obtained from samples from asymptomatic patients. It is known that such patients contribute significantly to the transmission of the virus, which means a higher viral load and, consequently, virus viability of these patients. Further investigation will clarify this issue and help to understand the value of antigen tests for the widespread detection of infectious agents in the context of the COVID-19 pandemic.

## Figures and Tables

**Figure 1 viruses-13-02012-f001:**
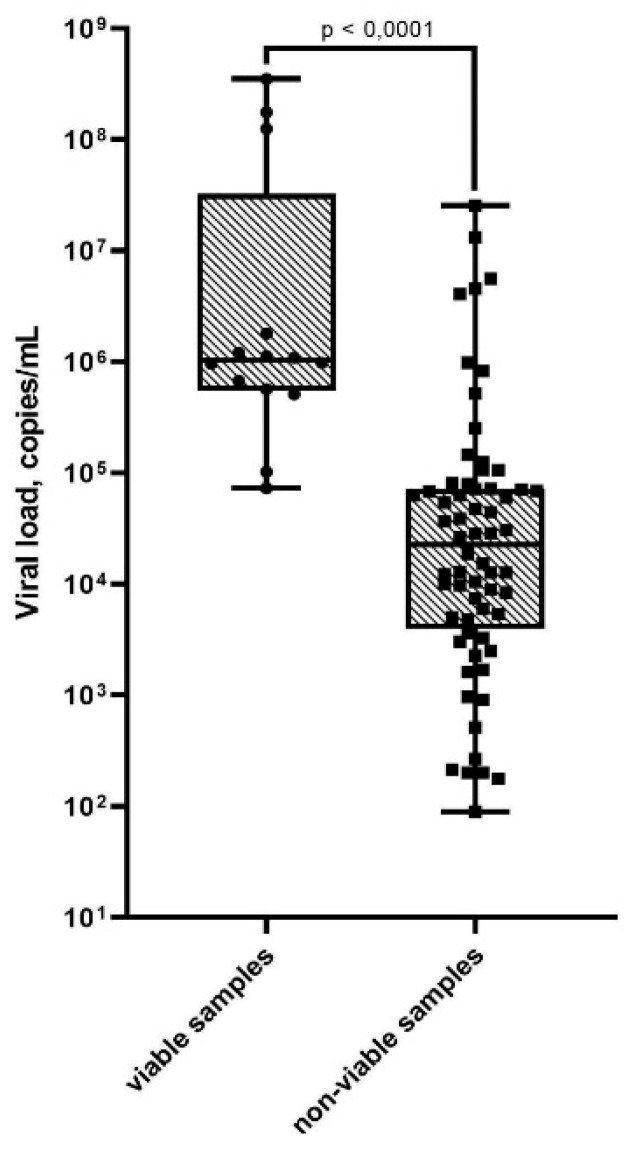
SARS-CoV-2 viral load in viable versus non-viable samples. Results are represented by a box plot: horizontal lines—medians; boxes—interquartile range; whiskers—min-max (*p*-value calculated using Mann–Whitney test).

**Table 1 viruses-13-02012-t001:** Assessment of sensitivity of rapid tests compared to RT-PCR with Alpha, Beta, and Delta variants of SARS-CoV-2.

Variants of SARS-CoV-2	Sensitivity		McNemar Test *p*-Value
	BIOCREDIT COVID-19 Ag	SGTi-Flex COVID-19 Ag	
B.1.1.1 EPI_ISL_421275	25.00%	33.33%	1000
B.1.1 EPI_ISL_1710865	50.00%	25%	0.248
B.1.1.7 EPI_ISL_1710866	58.33%	50.00%	1000
B.1.351 EPI_ISL_1257814	41.67%	25.00%	0.480
B.1.617.2	50.00%	58.33%	1000
EPI_ISL_2296305	33.33%	25.00%	1000
The Fisher’s Exact Test *p*-value	0.645	0.405	

**Table 2 viruses-13-02012-t002:** Assessment of sensitivity and specificity of rapid tests compared to RT-PCR.

SGTI-flex COVID-19 Ag			Biocredit COVID-19 Ag		
PCR result, *n*	negative	positive	PCR result	negative	positive
negative, *n* = 28	27	1	negative, n = 28	28	0
positive, *n* = 78	37	41	positive, n = 78	34	44
Sensitivity	52.56% (41/78)		Sensitivity	56.41% (44/78)	
Specificity	96% (27/28)		Specificity	100% (28/28)	

**Table 3 viruses-13-02012-t003:** Viral load (copies/mL) depending on viability and rapid test result.

Viral Load (Copies/mL)				
Quantitative Real-Time RT-PCR				
Value	Number of Positive Tests	Viral Load (Mean)	Range	Median
Successful isolation (viable virus)	14	4.71 × 10^7^	7.30 × 10^4^–3.50 × 10^8^	1.03 × 10^6^
Unsuccessful isolation (non-viable virus)	64	8.93 × 10^5^	89.00–2.56 × 10^7^	2.26 × 10^4^
BIOCREDIT COVID-19 Ag				
Successful isolation (viable virus)	14	4.71 × 10^7^	7.30 × 10^4^–3.50 × 10^8^	1.03 × 10^6^
Unsuccessful isolation (non-viable virus)	30	1.87 × 10^6^	89.00–2.56 × 10^7^	5.72 × 10^4^
RT-PCR “+”, antigen test “−” non-viable virus)	34	2.87 × 10^4^	1.77 × 10^2^–1.46 × 10^5^	9.85 × 10^3^
SGTI-flex COVID-19 Ag				
Successful isolation (viable virus)	11	5.97 × 10^7^	7.30 × 10^4^–3.50 × 10^8^	1.09 × 10^6^
Unsuccessful isolation (non-viable virus)	31	1.40 × 10^6^	89.00–2.56 × 10^7^	3.78 × 10^4^
RT-PCR “+”, antigen test “−” (non-viable virus)	37	4.87 × 10^5^	1.77 × 10^2^–1.31 × 10^7^	1.87 × 10^4^

**Table 4 viruses-13-02012-t004:** Assessment of sensitivity and specificity of rapid tests for viability.

	SGTI-Flex COVID-19 Ag		Biocredit COVID-19 Ag		Quantitative Real-Time RT-PCR	
	Positive	Negative	Positive	Negative	Positive	Negative
Viable virus	11	3	14	0	14	0
No viable virus	31	61	30	62	64	28
	Sensitivity 78.6% (49.2–95.3%) (11/14)	Specificity 66.3% (55.7–75.8%) (61/92)	Sensitivity 100% (76.8–100%) (14/14)	Specificity 67.4% (56.8–76.8%) (62/92)	Sensitivity 100% (76.8–100%) (14/14)	Specificity 30.4% (21.3–40.9%) (28/92)

## Data Availability

The data presented in this study are available in supplementary material here (Appendix A).

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
