# Peer review of "The Value of Rapid Antigen Tests for Identifying Carriers of Viable SARS-CoV-2"

_viruses, 2021, doi:10.3390/v13102012_

Round 1

Reviewer 1 Report

Comments on the manuscript ID viruses-1250728 on the value of rapid antigen tests to identify carriers of viable sars-cov-2 by Shidlovskaya E. V. et al., from Gushchin V. A. ’s lab.

Authors detected 106 hospitalized patients by using two quick antigen tests versus molecular technique: RT-PCR method and demonstrated the two Covid19 Ag tests (BioCredit and SGTI-flex) showed similar results in specificity and sensitivity. 14 isolates from the Antigen positive cases showed SARS-CoV-2 culture positive.

Convalescent sera from the 106 suspected Covid19 patients should be collected at about four weeks after onset of the symptom. In clinical diagnostic lab, the antibody response to SARS-CoV-2 should be detected by a sensitive or recognized method such as Abbott CoV-2 IgG kit.  All patients having increased antibody response against covid-19 could be named as positive group. Based on the serological diagnostic results authors could defined positive group, due to PCR assay can’t detect all infected cases but serological diagnosis can do.  

Authors did not describe if the Covid19 antigen tests can cross-react to other coronavirus viruses: 229E, OC43, NL63, HKU1 that cause common cold diseases.   Four tests might be enough for this purpose.

The Size of Figure 1 could be reduced.

Table 1 and table 3, Authors could improve it better, such as clearly indicate the calculation of the sensitivity and specificity as the sensitivity 52,56% (41/78). Specificity 96% (27/28).

Table 2, The first part of viral load (GE/ml) and quantitative real time RT-PCR viral isolates is unclear. I think authors means “viral isolates” and quantify the viral concentration by using qPCR assay. In this case, “quantitative real time RT-PCR” can be moved to figure legend where to explain how to calculate the among of the viruses.

 If the Ag tests are valuable for variants of SARS-CoV-2, such as UK variants: B.1.222; B.1.619; D614G; B.1.17.?

Supplement table 3 and 2 are interesting and present a lot of information. I suggest that the two tables could be presented in main text.

Minor question: Line 48: the “pilot” study: the “pilot” should be deleted.

In summary, authors reported the sensitivity and specificity of two antigens quick diagnostic methods. The data is interesting and methods is useful for the test at home such as point of care (POC).

Author Response

Comments and Suggestions for Authors

Comments on the manuscript ID viruses-1250728 on the value of rapid antigen tests to identify carriers of viable sars-cov-2 by Shidlovskaya E. V. et al., from Gushchin V. A. ’s lab.

Authors detected 106 hospitalized patients by using two quick antigen tests versus molecular technique: RT-PCR method and demonstrated the two Covid19 Ag tests (BioCredit and SGTI-flex) showed similar results in specificity and sensitivity. 14 isolates from the Antigen positive cases showed SARS-CoV-2 culture positive.

Convalescent sera from the 106 suspected Covid19 patients should be collected at about four weeks after onset of the symptom. In clinical diagnostic lab, the antibody response to SARS-CoV-2 should be detected by a sensitive or recognized method such as Abbott CoV-2 IgG kit.  All patients having increased antibody response against covid-19 could be named as positive group. Based on the serological diagnostic results authors could defined positive group, due to PCR assay can’t detect all infected cases but serological diagnosis can do. 

We agree that detecting antibodies in the serum of patients is a good way to identify convalescents.  But we would like to underline, the aim of our study was assessment the ability of antigen tests to detect patients shedding viable virus, but not as a diagnostic tool.

Authors did not describe if the Covid19 antigen tests can cross-react to other coronavirus viruses: 229E, OC43, NL63, HKU1 that cause common cold diseases.   Four tests might be enough for this purpose.

We did not test cross-reactivity to other coronavirus viruses: 229E, OC43, NL63, HKU1, because according to information of antigen tests manufactures the cross-reactivity was studied: 

  • BIOCREDIT COVID-19 Ag has been tested with 20 potentially cross reacting microorganisms and viruses. The results showed that BIOCREDIT COVID-19 Ag had no cross-reaction with microorganisms and viruses except very weak cross reacting with SARS-coronavirus.
  • SGTi-flex COVID-19 Ag was tested for potential crossreactivity using 33 samples containing antibodies to other pathogens and other states of disease (21 other viruses and 12 bacteria). No false positive results were observed with potential cross-reactants. https://www.diasys-diagnostics.com/fileadmin/downloads/study-reports/antigen/Analytical_performance_study_report__SGT_i-flex_COVID-19_Ag_R-LA-811-02_.pdf?_=1608218008

In addition, the absence of cross-reactivity to other coronavirus viruses: 229E, OC43, NL63, HKU1 was shown earlier for RapiGEN BIOCREDIT COVID-19 Ag test (https://www.thelancet.com/journals/lanmic/article/PIIS2666-5247(21)00056-2/fulltext)

The Size of Figure 1 could be reduced.

We agree with this comment to reduce the size of figure 1 in the manuscript. We revised the text of the manuscript and amended it.

Table 1 and table 3, Authors could improve it better, such as clearly indicate the calculation of the sensitivity and specificity as the sensitivity 52,56% (41/78). Specificity 96% (27/28).

We agree with this comment to improve Table 1 and table 3 in the manuscript. We revised the text of the manuscript and amended it.

Table 2, The first part of viral load (GE/ml) and quantitative real time RT-PCR viral isolates is unclear. I think authors means “viral isolates” and quantify the viral concentration by using qPCR assay. In this case, “quantitative real time RT-PCR” can be moved to figure legend where to explain how to calculate the among of the viruses.

In our study quantitative real-time RT-PCR was used as a method to evaluate viral load of samples collected from patients.  Table 2 shows a viral load and number of positive and negative samples tested by qRT-PCR and antigen tests with viable and non-viable virus.

If the Ag tests are valuable for variants of SARS-CoV-2, such as UK variants: B.1.222; B.1.619

; D614G; B.1.17.?

We agree that the evaluation of efficiency of antigenic tests with new variants of the virus is valuable information. Unfortunately, we did not have the opportunity to examine the efficiency of tests with variants B.1.222; B.1.619; D614G, but later the efficiency of  these Ag tests were verified with variants  B.1.1.1 (EPI_ISL_421275), B.1.1 (EPI_ISL_1710865), B.1.1.7 (EPI_ISL_1710866), B.1.351 (EPI_ISL_1257814), B.1.617.2  and hCoV-19/Russia/MOS-PMVL-1194/2021 (EPI_ISL_2296305). We added results of testing in the manuscript. 

Supplement table 3 and 2 are interesting and present a lot of information. I suggest that the two tables could be presented in main text.

We think to avoid overloading the main text, it would be better to keep Supplement table 3 and 2 in supplement.

Minor question: Line 48: the “pilot” study: the “pilot” should be deleted.

We agree with this comment. We revised the manuscript and deleted the “pilot”.

In summary, authors reported the sensitivity and specificity of two antigens quick diagnostic methods. The data is interesting and methods is useful for the test at home such as point of care (POC).

Reviewer 2 Report

This manuscript entitled “The Value of Rapid Antigen Tests to Identify Carriers of Viable SARS-CoV-2” by Elena V. Shidlovskaya and colleagues invested two Rapid Antigen Tests for their diagnostic value in identifying patients who excrete viable SARS-CoV-2.

 The authors showed ~50% sensitivity for both Tests from 106 patients, this really supersized me. As some other Tests in the market describe >95% sensitivity in their user instructions. Are the two Tests here just not good, can they represent other Antigen Test kit? What sensitivity level did these two kits offer on their own information pages? A well comparison with other kits in the market is necessary.

On the other hand, this study offered relation of viral load, virus isolation and test sensitivity. Interestingly, the sensitivity of antigen test is viral load dependent.

Another question is how much can the virus isolation represents the viable virus?

Last point, in table 3, the 28 RT-PCT negative tested patient, were they tested again in other time point? Also multi-time point test of the same patient with antigen test would be interesting data.

Author Response

This manuscript entitled “The Value of Rapid Antigen Tests to Identify Carriers of Viable SARS-CoV-2” by Elena V. Shidlovskaya and colleagues invested two Rapid Antigen Tests for their diagnostic value in identifying patients who excrete viable SARS-CoV-2.

 The authors showed ~50% sensitivity for both Tests from 106 patients, this really supersized me. As some other Tests in the market describe >95% sensitivity in their user instructions. Are the two Tests here just not good, can they represent other Antigen Test kit? What sensitivity level did these two kits offer on their own information pages? A well comparison with other kits in the market is necessary.

 BIOCREDIT COVID-19 Ag (RapiGEN Inc., Korea) and SGTI-flex COVID-19 Ag (Sugentech Inc., Korea)  tests are  widely used for point of care SARS-CoV-2  detection in Russia, so we chose these kits for our study. According to manufacturer's instructions  sensitivity of SGTI-flex COVID-19 Ag test is > 95.10% (CI 95%: 90.24%~97.61%), the specificity (negative agreement) is 99% (95% CI : 94.55%~99.82%), sensitivity of BIOCREDIT COVID-19 Ag is 92%. Unfortunately we did not use kits of other manufacturers in our study.

 On the other hand, this study offered relation of viral load, virus isolation and test sensitivity. Interestingly, the sensitivity of antigen test is viral load dependent.

According to data of other studies (https://onlinelibrary.wiley.com/doi/epdf/10.1002/jmv.27108, https://www.thelancet.com/journals/lanmic/article/PIIS2666-5247(21)00056-2/fulltext) efficacy of antigen tests depends on viral load.

Another question is how much can the virus isolation represents the viable virus?

In our study, successful virus  isolation is equal to a viable virus. Virus-induced cytopathic effect (CPE) was estimated  after 293T/ACE2 cell line infection by nasopharyngeal secretion (100 μL) from COVID-19 patients. Then, for samples with CPE after 5 days incubation real-time PCR was performed to confirm that CPE was caused by SARS-CoV-2 and not by other infectious agents that can cause CPE.

Last point, in table 3, the 28 RT-PCT negative tested patient, were they tested again in other time point? Also multi-time point test of the same patient with antigen test would be interesting data.

We did not test  patients with a negative PCR again in other time points. Based on our experience testing of hospitalized patients RT-PCT negative samples usually stay negative in later time points, this observation can be explained by the fact that patients are hospitalized at a relatively late stage of the disease, when the virus is no longer detected in nasal swabs.

Reviewer 3 Report

Major points of criticism:

  1. The aim of the study was to evaluate whether rapid antigen tests can be used to identify carriers of viable SARS-CoV-2. The underlying hypothesis could be (but is not specified in the manuscript) as follows: as antigen assays have a lower sensitivity compared to RT-PCR samples with a positive antigen test result (at least a high percentage of these samples) are expected to have sufficiently high viral loads to contain viable virus. The sensitivities and specificities of the tests for viral viability given in Table 3 are correctly determined. However, to address the aim of the study you would like to know how many of the antigen test ”+” samples have viable virus. This percentage is only 31,8% for the Biocredit COVID-19 Ag assay and 34.8% for the SGTI-flex COVID-19 Ag assay. Thus, the antigen tests investigated cannot reliably identify carriers of viable SARS-CoV-2. This result needs to be included in the “Results” and “Discussion” sections. The title must be adjusted as well.
  2. The introduction does not discuss why it is important to identify carriers of viable SARS-CoV-2. In view of the aim of the study, this points needs to be addressed.
  3. The sensitivities / specificities (relative to RT-PCR) of the Biocredit COVID-19 Ag and SGTI-flex COVID-19 Ag tests reported by the manufacturers are 90.2% / 100% and 95.1% / 99%, respectively. The sensitivities of the two antigen tests reported in the present study (i.e., 56.41% and 52.56% for the Biocredit COVID-19 Ag and SGTI-flex COVID-19 Ag test, respectively; Table 1) are thus far lower than expected. This should be discussed.
  4. Comparison of sensitivity and specificity between the two antigen tests and RT-PCR was done on “three nasopharyngeal samples from each patient, two of which were used for antigen testing.” (p. 2, ll.65-66). It is known that different swab samples from the same patient, even if collected directly one after the other, can yield varying viral loads. This problem should be addressed in the “Discussion” section of the manuscript.
  5. It seems (but is not absolutely clear) that the third sample that “was transferred to tubes with 1 mL of phosphate buffered saline (PBS).” (p. 2, l. 66) was the one that was used for RT-PCR. Swabs used for SARS-CoV-2 testing are usually transferred to tubes with virus transport medium (VTM) or universal transport medium (UTM). The fact that only PBS was used instead of a medium may considerably lower the amount of viable virus in the samples.
  6. It is not clear whether virus isolation was performed only with the samples that were transferred to tubes with PBS after collection or with all three samples. This needs to be clarified in the “Materials and methods” section.
  7. The terms sensitivity and specificity are used in relation to two different test systems, i.e., RT-PCR and viral viability. It should be clear which relative sensitivity / specificity is meant each time the terms are used, in particular in the results part of the abstract.
  8. 5, ll. 158-160: “Rapid tests were able to give a positive result on samples with median values of 5.72×104 and 3.78×104 (GE/mL) for Biocredit COVID-19 Ag and SGTI-flex COVID-19 Ag, respectively.” However, these median values are shown in Table 2 for “Unsuccessful isolation”!?
  9. Sensitivity and specificity should be specified in %.
  10. There is a discordance between Table 2 and Table 3: the number of samples tested positive with the SGTI-flex COVID-19 Ag test, but without viable virus is specified as “30” in Table 2, whereas this number is “31” in Table 3.

Minor points of criticism:

  1. The paragraph “SARS-CoV-2 testing” in the “Materials and methods” section should read “RT-PCR” (as antigen testing is also SARS-CoV-2 testing).
  2. 2, line 52: “…who had undergone two rapid tests: quantitative RT-PCR and viability assessment…” should read as follows “…who had undergone two rapid tests, quantitative RT-PCR and viability assessment…”.
  3. In the “Materials and methods” section, “SARS-CoV-2 testing”, a reference with the number “29” is given: “The protocol for qPCR-RT used in this study had been described previously [29].” (p. 2, l. 84). This reference does not appear in the list of references.
  4. For virus isolation, “…nasopharyngeal secretion (100 μL) from COVID-19 patients was added to tablets.” (p. 2, ll.93-94). What does “tablets” mean?
  5. “smears” (p. 3, l.117) should read “swabs”.
  6. Define “GE/mL” (p. 3, line 119) (= genomic equivalents) and/or use “copies/ml” for viral load as in Suppl. Table 1.
  7. 5, ll. 165-166: “Overall, out of 106 samples, viable virus was detected in 14 patients, representing 13.2% of all participants.” The percentage of samples with viable virus should be given in reference to the number of RT-PCR-positive samples, not the total number (N= 106) of samples.

Author Response

Major points of criticism:

  1. The aim of the study was to evaluate whether rapid antigen tests can be used to identify carriers of viable SARS-CoV-2. The underlying hypothesis could be (but is not specified in the manuscript) as follows: as antigen assays have a lower sensitivity compared to RT-PCR samples with a positive antigen test result (at least a high percentage of these samples) are expected to have sufficiently high viral loads to contain viable virus. The sensitivities and specificities of the tests for viral viability given in Table 3 are correctly determined. However, to address the aim of the study you would like to know how many of the antigen test ”+” samples have viable virus. This percentage is only 31,8% for the Biocredit COVID-19 Ag assay and 34.8% for the SGTI-flex COVID-19 Ag assay. Thus, the antigen tests investigated cannot reliably identify carriers of viable SARS-CoV-2. This result needs to be included in the “Results” and “Discussion” sections. The title must be adjusted as well.

We obtained 31,8% (14/44) “+” antigen test whis viable virus for the Biocredit COVID-19 Ag assay and 26.2% (11/42) for the SGTI-flex COVID-19 Ag assay. This is not much, but for RT-PCR it is only 17,9% (14/78). So, we are concerned that studying tests are not worse than RT-PCR assay. We compared these assays with RT-PCR by chi-square test and got p =  0.5039 and p = 0.4095 for  Biocredit COVID-19 Ag assay and for the SGTI-flex COVID-19 Ag assay respectively. P-value between two rapid antigen tests is equal to 0.7361. We added this to the “results” and “discussion”  sections.

  1. The introduction does not discuss why it is important to identify carriers of viable SARS-CoV-2. In view of the aim of the study, this points needs to be addressed.

We agree with this comment. We revised the manuscript and added  importance  of identification carriers in Introduction.

  1. The sensitivities / specificities (relative to RT-PCR) of the Biocredit COVID-19 Ag and SGTI-flex COVID-19 Ag tests reported by the manufacturers are 90.2% / 100% and 95.1% / 99%, respectively. The sensitivities of the two antigen tests reported in the present study (i.e., 56.41% and 52.56% for the Biocredit COVID-19 Ag and SGTI-flex COVID-19 Ag test, respectively; Table 1) are thus far lower than expected. This should be discussed.

The results of our study are close to the previously published Cochrane meta-analysis, according to which sensitivity of Ag tests was from 58.1% to 79.0% in different groups of participants. Also, according to same Cochrane meta-analysis sensitivity was high in those with cycle threshold (Ct) values on PCR ≤25 (94.5%, 95% CI 91.0% to 96.7%; 36 evaluations; 2613 cases) compared to those with Ct values >25 (40.7%, 95% CI 31.8% to 50.3%; 36 evaluations; 2632 cases). In our study, we determined the sensitivity for all samples without cutoff by Ct values. This is why sensitivity of the Biocredit COVID-19 Ag and SGTI-flex COVID-19 Ag test is lower than expected.

  1. Comparison of sensitivity and specificity between the two antigen tests and RT-PCR was done on “three nasopharyngeal samples from each patient, two of which were used for antigen testing.” (p. 2, ll.65-66). It is known that different swab samples from the same patient, even if collected directly one after the other, can yield varying viral loads. This problem should be addressed in the “Discussion” section of the manuscript.

We agree with this comment. We revised the manuscript and added  this part in the  “Discussion” section of the manuscript.

  1. It seems (but is not absolutely clear) that the third sample that “was transferred to tubes with 1 mL of phosphate buffered saline (PBS).” (p. 2, l. 66) was the one that was used for RT-PCR. Swabs used for SARS-CoV-2 testing are usually transferred to tubes with virus transport medium (VTM) or universal transport medium (UTM). The fact that only PBS was used instead of a medium may considerably lower the amount of viable virus in the samples.

In our study, samples of nasopharyngeal secretions were used for both quantitative RT-PCR and viral isolation. Accordingly, we could not rule out the influence of the multicomponent composition of the viral transport medium (VTM) on RNA quality. In addition, recent studies show that the detection of inactivated SARS-CoV-2 in samples of nasopharyngeal secretion placed in PBS or VTS is equivalent and the virus remains stable (https://www.ncbi.nlm.nih.gov/pmc/articles/ PMC7492856 /; https://www.ncbi.nlm.nih.gov/pmc/articles/PMC7219422/; https://journals.asm.org/doi/10.1128/JCM.01094-20). Another study also demonstrated that there is no significant difference in viral yield from different swabs and most vehicles for collection and subsequent detection of SARS-CoV-2 (https://www.sciencedirect.com/science/article/pii/S0166093420301993; https://www.ncbi.nlm.nih.gov/pmc/articles/PMC7269412/). Thus, it is expected that the use of a PBS-based transport medium may have a minimal impact on the success rate of SARS-CoV-2 isolation. It can also be emphasized that, according to numerous studies, in liquids SARS-CoV-2 can remain infectious for an extended period, while the samples in this study were exposed to minimal environmental exposure.

  1. It is not clear whether virus isolation was performed only with the samples that were transferred to tubes with PBS after collection or with all three samples. This needs to be clarified in the “Materials and methods” section.

We revised the manuscript and clarified the “Materials and methods” section. We added “The first sample was transferred to tubes with 1 mL of phosphate buffered saline (PBS) and then used for RT-PCR and virus isolation. On the next day (after getting results of RT-PCR), participants were tested with antigenic tests.” in the  “Sample collection and transportation” section of the manuscript.

  1. The terms sensitivity and specificity are used in relation to two different test systems, i.e., RT-PCR and viral viability. It should be clear which relative sensitivity / specificity is meant each time the terms are used, in particular in the results part of the abstract.

We agree with this comment. We revised the manuscript and clarified the results part of the abstract.

  1. 5, ll. 158-160: “Rapid tests were able to give a positive result on samples with median values of 5.72×104 and 3.78×104 (GE/mL) for Biocredit COVID-19 Ag and SGTI-flex COVID-19 Ag, respectively.” However, these median values are shown in Table 2 for “Unsuccessful isolation”!?

We agree with this comment. We revised the manuscript and amended the Table 3  (we added new table in the text, so Table 3 used to be Table 2 in  the previous version of the manuscript)

  1. Sensitivity and specificity should be specified in %.

We agree with this comment. We revised the manuscript and specified sensitivity and specificity in %.

  1. There is a discordance between Table 2 and Table 3: the number of samples tested positive with the SGTI-flex COVID-19 Ag test, but without viable virus is specified as “30” in Table 2, whereas this number is “31” in Table 3.

We agree with this comment. We revised the manuscript and fixed this mistake.

Minor points of criticism:

  1. The paragraph “SARS-CoV-2 testing” in the “Materials and methods” section should read “RT-PCR” (as antigen testing is also SARS-CoV-2 testing).

We agree with this comment. We revised the manuscript and “quantitative RT-PCR”  in the “Materials and methods” section.

  1. 2, line 52: “…who had undergone two rapid tests: quantitative RT-PCR and viability assessment…” should read as follows “…who had undergone two rapid tests, quantitative RT-PCR and viability assessment…”.

We agree with this comment. We revised the manuscript and fixed this mistake.

  1. In the “Materials and methods” section, “SARS-CoV-2 testing”, a reference with the number “29” is given: “The protocol for qPCR-RT used in this study had been described previously [29].” (p. 2, l. 84). This reference does not appear in the list of references.

We agree with this comment. We revised the manuscript and fixed this mistake.

  1. For virus isolation, “…nasopharyngeal secretion (100 μL) from COVID-19 patients was added to tablets.” (p. 2, ll.93-94). What does “tablets” mean?

We agree with this comment. We revised the manuscript and fixed this mistake: “tablets” is a cell of 96-well plate. We fixed this mistake.

  1. “smears” (p. 3, l.117) should read “swabs”.

We agree with this comment. We fixed this mistake.

  1. Define “GE/mL” (p. 3, line 119) (= genomic equivalents) and/or use “copies/ml” for viral load as in Suppl. Table 1.

We agree with this comment. We revised the manuscript and used “copies/ml” for viral load in in the whole article.

  1. 5, ll. 165-166: “Overall, out of 106 samples, viable virus was detected in 14 patients, representing 13.2% of all participants.” The percentage of samples with viable virus should be given in reference to the number of RT-PCR-positive samples, not the total number (N= 106) of samples.

We revised the manuscript and added in sentence: “Overall, out of 106 samples, a viable virus was detected in 14 patients, representing 13.2% of all participants and 17,9% of RT-PCR positive participants”.

Round 2

Reviewer 1 Report

The manuscript has been improved. However, the important information of convalescent sera from 106 cases did not present in the revised manuscript. The two methods for SARS-CoV-2 antigen tests are not sufficient to define which patients are real infection or nonspecific reaction. From clinical point view, increased SARS-CoV-2 antibody titter is the only proof for supporting  the antigen test methods. 

Author Response

The manuscript has been improved. However, the important information of convalescent sera from 106 cases did not present in the revised manuscript. The two methods for SARS-CoV-2 antigen tests are not sufficient to define which patients are real infection or nonspecific reaction. From clinical point view, increased SARS-CoV-2 antibody titter is the only proof for supporting  the antigen test methods.

We revised the text of the manuscript and added information concerning evaluation of convalescent sera from patients.

Reviewer 2 Report

Table 3 in Supplementary need some detail information

I suggest to cite a recent article' In Vitro Rapid Antigen Test Performance with the SARS-CoV-2 Variants of Concern B.1.1.7 (Alpha), B.1.351 (Beta), P.1 (Gamma), and B.1.617.2 (Delta'

Author Response

Table 3 in Supplementary need some detail information

We agree with this comment to provide more detail information in table 3 of Supplementary. We revised the table and amended it.

I suggest to cite a recent article' In Vitro Rapid Antigen Test Performance with the SARS-CoV-2 Variants of Concern B.1.1.7 (Alpha), B.1.351 (Beta), P.1 (Gamma), and B.1.617.2 (Delta'

We agree with this comment and cited recommended article.